# Evaluation of Linguistics Students' Learning Outcomes in Peer Teaching Courses: The Effect of Altruistic and Egoistic Behaviors

Natalia Anosova, Aleksandra Dashkina, Aleksandra Kobicheva *, Ekaterina Shostak and Dmitriy Tarkhov

Institute of Humanity, Peter the Great Saint-Petersburg Polytechnic University, 195251 St. Petersburg, Russia; anosova_ne@spbstu.ru (N.A.); dashkina_ai@spbstu.ru (A.D.); shostak_ev@spbstu.ru (E.S.); tarkhov_da@spbstu.ru (D.T.)
* Correspondence: kobicheva_am@spbstu.ru

**Abstract:** In the current study, we evaluated the students' foreign language lexical and grammatical skills in the course based on the peer teaching methodology and analyzed the effect of their altruistic and egoistic behaviors on learning results. This experiment was conducted in a groups of senior students majoring in linguistics. The total number of participants accounted for 197 students (101 students in reference groups and 96 in exposure groups); the difference between the reference and exposure groups was that the undergraduates in the latter were to prepare a fragment of a lesson, create exercises, and act in the capacity of a teacher during the course. To evaluate students' foreign language lexical and grammatical skills, the diagnostic test was conducted at the beginning and at the end of the experiment. Apart from comparing the diagnostic and final tests, we also circulated a questionnaire which checked the students' egoistic and altruistic tendencies. The data appeared to be quite noisy; therefore, we processed them with a tool which proves effective when it comes to solving such problems, i.e., neural networks. According to the results on learning outcomes, students improved their English proficiency in the exposure groups to a greater extent than in the reference groups. At the same time, the results of the psychological tests revealed that the higher the students' training level, the less altruistic they are. Also, it was detected that the more altruistic learners' progress in outcomes was higher than those of the more selfish students, regardless of the way in which the learning process was organized. Moreover, the statistical data proved the efficiency of the peer teaching methodology for students' majoring in linguistics, despite their psychological characteristics.

**Keywords:** English-as-a-foreign-language (EFL); peer mentoring programs; altruistic and egoistic behaviors; lexical and grammatical skills; learning performance; neural network

## 1. Introduction

In the context of rapidly changing conditions and the socializing factors of modern society, fundamental values, traditional ideas, social stereotypes, and moral guidelines are being transformed. Changes also concern social ideas about altruism and egoism and the need for social behavior or individualistic attitudes, and there is a contradiction between altruistic and egoistic aspirations of people. On the one hand, the role of helping behavior increases due to the increasing number of those who need help and support. On the other hand, competition and the need for personal well-being lead to an increase in individualistic attitudes, an increase in indifference and selfishness, and a decrease in the desire to help. The socio-psychological phenomena of altruism and egoism have different causes and manifestations, as well as different external and internal factors that strengthen or restrain their manifestations and consequently contribute to or hinder the process of learning, thus affecting the learning outcomes.

Some research has been conducted on the correlation between students' personality traits and their learning capacity [1–3]. In particular, different types of assessment (exams, writing assignments, test, etc.) were analyzed in terms of their efficiency for different personality types (sensitive people (blue); detail-oriented and prepared (gold); calm in tense

situations (green); and adventurous people who take risks (orange)) [1]. The correlation between the individual's change orientation (the way a person thinks about change) and the individual's learning orientation (instruction-oriented learning behavior, planned learning behavior, meaning-oriented learning behavior, and emergent learning behavior) has been studied [2]. Also, the correlations between personality traits (warmth, self-reliance, openness to change, etc.), cognitive academic competences(analyzing, focus, etc.), and academic outcomes have been determined [3]. As can be seen, no research has been conducted on the study of the correlation between learning behaviors and altruistic/egoistic behaviors (as personal traits).

Foreign language learners often experience a so-called plateau effect—they feel that they are stuck and have stopped making any progress; they no longer incorporate new vocabulary or complex grammatical structures in their speech, so their language proficiency level does not improve whatsoever. It is often caused by their lack of independence from the teacher, who makes decisions, sets objectives, and facilitates the learning process. One of the ways to mitigate the plateau effect is students' intense involvement in classroom activities. It can be achieved through getting them to participate in teamwork, research projects, discussions, and competitions and having them act in the capacity of a teacher.

When students run a fragment of a class for their groupmates, they cannot but become deeply involved in the learning process. First of all, when they teach their peers, they need to be word-perfect and learn as much additional material as possible, otherwise they will not be able to explain the topic properly or answer their groupmates' questions. It means that the students who perform as a teacher broaden their knowledge even at the point of preparing a fragment of a class. On top of that, they improve their language skills when they ask their peers questions, correct their mistakes, and respond to their queries. Another factor that contributes to the acquisition of foreign language skills is frequent repetition of the learning material in the course of running a fragment of a class.

The aim of the current study is to evaluate the students' foreign language skills in the peer teaching course and analyze the effect of students' altruistic and egoistic behaviors on students' learning outcomes.

Specifically, this study is focused on three major research questions:

1. Does the peer teaching methodology contribute to linguistics students' higher learning outcomes?
2. Do altruistic and egoistic behaviors influence students' learning outcomes?
3. How do both the initial level of foreign language training and the psychological characteristics influence the learning outcomes?

This paper starts by introducing the theoretical background on the factors influencing learning performance, altruistic and egoistic behaviors of students, and their influence on different educational methods' efficiency. Section 1 analyses the factors influencing learning performance and describes the studies on peer mentoring experiences as well as the influence of altruistic and egoistic behaviors on students' performance. Section 2 presents the research sample, the lexical and grammatical skills questionnaire, altruism and egoism scales, and also the conducted analysis of research variables. In Section 3, we determine the learning performance indicators for all groups of students and explore the impact of altruism and egoism scales on students learning results. In Section 4, the research results are demonstrated and discussed. Finally, it concludes with limitations and recommendations for future researchers and practitioners.

*1.1. Literature Review*

1.1.1. Factors Influencing Learning Performance

One of the major factors that influence students' academic performance is the degree of their involvement in classroom activities. Even the learners who are reluctant to collaborate with their peers should be plunged into meaningful classroom and out-of-class activities. This can be achieved by giving each undergraduate a role which will make him/her feel valued and appreciated by their peers [4,5]. Having learners act in the capacity of a teacher

can be regarded as an example of such a meaningful activity. Students can either give a fragment of a class to their peers or act as tutors in the course of teamwork or pair work.

If learners are engaged in classroom activities, they are more motivated as they are given more learning autonomy. Their learning outcomes are much better than in a conventional class, in which they have to work with dull and meaningless exercises from their course book [6]. The students who act as teachers are more motivated since they feel in the position to control their groupmates and share their expertise with the latter. On the other hand, the other learners understand explanations better when they are given by a person of the same age who uses similar vocabulary and speaks in a simple and comprehensible language than they do in the conventional classroom environment.

Peer mentoring helps the students who act in the capacity of a teacher to acquire interpersonal skills which may prove useful when they embark on their career. A higher degree of personal accountability is one of the essential soft skills that learners develop when they have to assume responsibility for the quality of the knowledge that they share with their peers. Learners who participate in peer mentoring programs are more likely to complete their degree course [7]. By taking on the teacher's role, students turn from detached observers into active participants of the educational process, which will generate strong motivation regardless of the form that peer mentoring might take place. For example, if learners are regularly given an assignment to prepare and have to teach a fragment of a class, they are involved in active cognitive activities. Perhaps this is one of the reasons why mentors are less likely to drop out.

In the experiment conducted at the University of Ruse (Bulgaria), a number of students were given the opportunity to teach a fragment of a class, and afterwards, the instructor compared the learning outcomes in the exposure groups with those in the reference ones and circulated a questionnaire in which the subjects were asked to comment on their experiences. The outcomes in the exposure groups were considerably higher than in the reference ones. The students of the exposure groups appeared to be more motivated and interested in learning, as well as more persistent when they had to complete challenging tasks. Peer teaching provided the students with new insights into how to learn and into the subject matter of the learning content [8–11]. Thus, peer teaching in an EFL classroom can be regarded as a kind of controlled practice since in the course of preparing a fragment of a class, students delve deeply into the learning material and revise it thoroughly to give their groupmates clear and well-structured explanations.

When students assume the teacher's role, apart from explaining new material to their peers and checking their understanding, they also assess the other learners' progress. This activity helps them develop critical judgment and improves their knowledge of the subject [12]. For example, when one of the learners is given an assignment to check other students' essays, he often has difficulty working out if a sentence contains a lexical or grammatical mistake, so he has to use reference materials, such as course books or online dictionaries. In this way, the learner acting in the capacity of the teacher revises grammar and vocabulary.

In general, peer teaching has a number of advantages. Students who assume the teacher's role are the same age as the audience they work with, so they know what problems their peers are likely to face. On top of that, the atmosphere in a student-led classroom is more informal and relaxed than in a conventional one, which is conducive to group interaction [13]. Given the considerable advantages of peer teaching, students should regularly be given the opportunity to act in the capacity of a teacher since it will improve their knowledge of the subject, upgrade their interpersonal skills, and encourage them to take a vital part in classroom activities.

Since the classroom atmosphere is more relaxed when a student acts in the capacity of a teacher, peer teaching also alleviates students' anxiety by raising their self-awareness and self-esteem; they no longer feel isolated from the rest of the group. Moreover, it helps students take responsibility for their own learning and lowers dropout rates [14]. Thus, in order to form essential skills and feel part of a close-knit learning community,

students should take turns teaching their peers. Owing to tremendous rapport with their groupmates and deep involvement in classroom activities, even mediocre students become more enthusiastic and motivated.

### 1.1.2. Peer Mentoring and Motivation, Altruism vs. Egoism

There is some research based on computational methods that investigates the role of altruistic strategies in group survival. Mostly, the results demonstrate that public-oriented behavior outperforms self-oriented behavior in the long run and gives more benefits for adaptiveness and survival [15]. However, other computational research comparing selfish learning (each agent aims to increase an individual reward) to sequential social learning (agents learn from their neighbors) denotes that the former learning style contributes to better accuracy [16]. Thus, we can assume that both strategies are beneficial if applied properly and with respect to the desired outcome—overall group success or better accuracy (learning result). Further research is needed to investigate these two strategies in language learning.

The existing literature states a significant positive effect of reciprocal teaching strategies on student motivation. Students instructed with such strategies showed not only more interest in performing the task but also expressed more enjoyment from the learning process in comparison with students from the control group who were instructed with traditional, nonreciprocal strategies. In turn, on a larger scale, it encourages students to enjoy the subject [17]. Peer mentoring also promotes active interaction between students and is deemed as a success factor for boosting confidence on an individual level [18].

Reciprocal teaching (that goes along with altruistic strategies) is also shown to improve student performance at the individual level (the study was focused on reading comprehension), which contributes to developing communicative competence and encourages group work [19], in turn, increasing motivation and learning process satisfaction.

What makes students become peer mentors? The need for power is prevalent, followed by the need for affiliation and the need for achievement [20]. Thus, self-oriented motives such as the need for experiencing personal power play a role in making people work for the common benefit.

Another factor that motivates people to use altruistic strategies is empathy. People, in general, are more inclined to give assistance to a person whom they consider similar to themselves, especially when this similarity is additionally highlighted or pointed out [21]. In an educational context, such altruistic behavior can be induced by underlining common values or a common situation that students encounter while studying, thus evoking empathy in them.

What is more surprising is that empathy drives altruistic behavior, even in people with more egoistic motives. A person wanting to feel good after helping (reward anticipation) and wanting to avoid negative emotions for not helping (fear of social condemnation) will most likely go for the first option. There is another study which reveals that subjects with egoistic motives might show altruistic behavior if it is proven or at least subjectively (due to culture factors) seems to be the best way to gain benefits individually [22].

Cooperation and language learning have much in common: they are considered to be essential for being human, their evolution is thought to be a puzzle or a mystery, and it is difficult to explain their emergence without assuming that humans developed these traits to survive as a species or "for the good of the group" [23–25]. It is evident that cooperation and altruism can benefit the group, whereas it is much less clear how they benefit individual performance. The same is true for language. Linguistic communication can be used to share knowledge, to exclude cheaters by spreading gossip about them, and to plan well-coordinated actions [26–28]. All of this would give an advantage to the group. At the same time, it is well known that complex traits cannot evolve simply because they benefit the group or the individual [29–31].

The term altruism was first used by Auguste Comte [32]. It is one of the few terms born within the scientific field that will enter the common language roughly maintaining

the same meaning. For the positivist Comte, altruism represented the powerful impulse to the intellectual and moral development of humanity.

To some extent, altruism merges with egoism.

The link between egoism and altruism may therefore be identified as a rational form of benevolence through which a person can be empowered by aiding others in their striving. Spinoza scholar Steven Nadler explains that a virtuous person will treat others in such a way that their own performance is increased [33]. Spinoza's conception of egoism clearly illustrates how egoism and altruism (without pity) collapse into the same thing and it also motivates the reason for which we need to form collective concepts of the good so that we can strive for the same things without posing a threat to one another [34].

Despite the fact that most people understand altruism as selfless helping behavior towards other people or society as a whole, sometimes contrary to their own interests, there are many nuances and contradictions in the understanding of this socio-psychological phenomenon. In today's dynamically changing society, social ideas about altruism and egoism are also undergoing changes, so it is important to study the attitude of modern youth to altruistic manifestations, to social behavior, or to individualistic attitudes and, accordingly, to the egoistic behavior model [35].

Within the framework of different disciplinary approaches, preference is given either to the evolutionary, genetic interpretation of the origin and development of altruism [36] or to the emphasis on the social aspect when explaining the selfless mutual assistance of strangers and understanding altruism not as a contradictory phenomenon but rather a complementary one [37]. The correlation of motives in altruistic or egoistic behavior in situations of social interaction is complex and interdependent. A person can help another person at a specific moment selflessly, not counting on a return service, but at the same time, subconsciously focusing on maintaining or establishing interpersonal relationships that will help meet any needs in the hypothetical future. Therefore, the ability to communicate positively in the community allows a person who commits altruistic acts to feel satisfied, without paying attention to the immediate benefits. Thus, social behavior is based not only on altruistic but also on egoistic motives. A person who helps another person counts on reciprocity, i.e., support, if not now, then in the future, if necessary, to satisfy their material or psychological needs [38]. This point of view is quite common, and theories are being developed that prove the dual motivation of prosocial behavior. In addition, active social interaction encourages altruistic behavior and limits egoistic behavior [39].

## 2. Materials and Methods

### 2.1. Participants

The experiment was conducted with groups of senior students majoring in linguistics. The total number of participants comprised 197 students (101 students in reference groups and 96 in exposure groups), specializing in pedagogy, translation, and teaching Russian as a foreign language. The students in all of the groups participating in the experiment worked with the same course book *Upstream Proficiency*. The difference between the reference and exposure groups was that the undergraduates in the latter were to prepare a fragment of a lesson, create exercises, and act in the capacity of a teacher during the course. The experiment was conducted in the fall semesters of 2021 and 2022 (14 weeks each semester). At the beginning of the experiment, we put forward a hypothesis that the peer teaching approach would be conducive to more successful learning outcomes than those that can be produced in a conventional classroom environment. We also formulated another hypothesis about the positive correlation between the levels of learners' altruism and their learning outcomes.

### 2.2. Description of Educational Methods

In both the reference and exposure groups, each class had a similar structure. It started with a 5 min warm-up exercise, in which the students were given a discussion point. For example, in unit 4 ("The happiest days of your life?"), the students were asked

some questions about the school subjects they had had at school and the courses they had completed at university so that they could focus on the topic "Education". Then, they completed a short listening comprehension exercise, which also took five minutes. After that, they answered the questions based on the text from the course book which they had read as part of their home assignment. The questions either covered the content of the text or were aimed at eliciting the students' opinion about its main ideas. The teacher also had the learners translate 10 sentences which contained the new words from the text from Russian into English. The question and answer session and the translation took 15 min. The next step was a 10 min listening and speaking exercise from the course book (they were to listen to a short audio record, complete a multiple-choice exercise which checked their understanding, and discuss some points related to this record using at least 5 new words from the text). The listening and speaking sections were followed by two vocabulary exercises. The students were to fill out the gaps with new words, choose the correct word to complete the sentences, or complete a word formation exercise. Then, the teacher asked the students 10 questions, the answers to which were to include one of the new words. This vocabulary practice took another 15 min. At the next stage, the students were asked to do 4 grammar exercises from the course book to revise the material they had studied before. The teacher also helped them consolidate this material by asking them some questions or having them continue the sentences with their own ideas. The grammar section of the class also took 15 min. For the rest of the class (30 min), the students worked with an additional topic-orientated book [40] in which they were given a maxim or a catchphrase with a number of leading questions, extracts from various internet resources which illustrated the point made by the author of the quotation, and a vocabulary list related to the topic. The students were to comply with the key requirement: they had to use at least 15 new words either from the vocabulary list in the additional textbook or from the corresponding unit in *Upstream Proficiency* as well as grammatical structures that were revised in the course of working with the grammar section.

In the reference groups, the teacher organized and facilitated all of the classroom activities and kept track of the time, whereas in the exposure groups, the students were given an opportunity to run fragments of either the first or the second half of the class. For example, one student facilitated the discussion in the warm-up exercise, another student asked the questions about the content of the text and asked his peers to translate the sentences which contained the words from the text into English (the questions and the sentences had been prepared by him in advance), and the third student supervised his peers' listening comprehension assignment and the discussion based on the listening excerpt. Another student oversaw the lexical exercises from the course book and then had his peers consolidate the new words by asking them pre-prepared questions. The second part of the class was conducted by the teacher. If the teacher ran the first part of the class, the learners were to supervise the second half of it. One student superintended the revision of grammar. He asked them some questions that he had prepared in advance or gave them the beginning of the sentences which they continued with their own ideas. The purpose of this exercise was to have them use the grammatical structures they had just revised. Two students worked in turns with the additional textbook (for 15 min each). They facilitated the discussion and added some factual information if the other learners could not come up with relevant ideas to support their arguments. Before the class, they familiarized themselves with some additional materials from various internet resources so they were in the position to keep the discussion on track. The classroom activities that the students were in charge of changed throughout the experiment; for example, if a student supervised the listening comprehension section, next time, he oversaw the vocabulary fragment of the class.

### 2.3. Measures

To evaluate students' foreign language lexical and grammatical skills, the diagnostic test was conducted at the beginning of the experiment. It was a multiple-choice test which

contained 30 sentences (15—to check grammar, 15—to check vocabulary); the maximum score was 30. We used multiple-choice tests as the most relevant tool for assessing the learners' language proficiency for several reasons. The tests were developed in such a way that the students were unlikely to be able to guess the correct answers. Moreover, anecdotal evidence suggests that the students who achieve high test scores also speak English more fluently with fewer mistakes. The test was aimed at gaging the initial level of the participants' foreign language proficiency, and the assignments were based on the course book they had completed in their junior year. The examples of the sentences from the test are given below:

Vocabulary: I shouldn't have talked to you about the things that are deeply personal, but now I am glad I . . .

A.    set it from my mind.
B.    put it out of my soul.
C.    let it away from my heart
D.    got it off my chest.

Grammar: I revised all the grammar we had studied since the beginning of our course, but I . . . it because the test covered only the last semester.

A.    didn't need to do
B.    needn't have done
C.    didn't have to do
D.    mustn't have done

The average result in the reference groups was 21.18 in 2021 and 21.04 in 2022; the average result in the exposure groups was 20.95 in 2021 and 21 in 2022.

At the end of the fall semesters in 2021 and 2022, all of the participants wrote a final test. It was in the same format as the diagnostic one, and it was aimed at checking the grammar and vocabulary that they had studied in the course of the experiment, so it included the material both from *Upstream Proficiency* and from the additional course book.

Vocabulary: Currently, in medical circles, vitamin D is the . . .

A.    topic on the tongues.
B.    light in all windows.
C.    talk of the town.
D.    chip on the shoulder.

Grammar: No sooner . . . pleasantries . . . down to business.

A.    we exchanged/did we get
B.    had we exchanged/than we got
C.    had we exchanged/when we got
D.    we exchanged/then had we got

Apart from comparing the diagnostic and final tests, we also circulated a questionnaire which checked the students' egoistic and altruistic tendencies.

In our study, we used a test from the course book by Fetiskin, N.P., Kozlov V.V., and Manuilov "Socio-psychological diagnostics of personality development and small groups" [41]. The test is aimed at determining an individual's personal socio-psychological orientations and identifying altruistic attitudes. It included the following questions:

1.    Do people often tell you that you think more about others than about yourself?
2.    Is it easier for you to ask for others than for yourself?
3.    Is it difficult for you to say "no" to people when they ask you for something?
4.    Do you often try to help people if they are in trouble?
5.    Do you prefer to do something for yourself rather than do something for other people?
6.    Do you try to do as much as possible for other people?
7.    Do you believe that the greatest value in life is to live for other people?
8.    Is it difficult for you to make yourself do something for others?

9.　　Is selflessness one of your personality traits?
10.　Are you convinced that caring for others is often to your own detriment?
11.　Do you disapprove of people who cannot take care of themselves?
12.　Do you often ask people to do something for selfish reasons?
13.　Is the desire to help other people one of your personality traits?
14.　Do you think that a person should think about himself first, and then about others?
15.　Do you usually devote much time to yourself?
16.　Do you believe that it is not worth going to great lengths to help other people?
17.　Do you usually have neither the energy nor the time for yourself?
18.　Do you spend your free time only pursuing your hobbies?
19.　Can you describe yourself as a selfish person?
20.　Are you willing to take maximum efforts only for a good reward?

A respondent gets one point for each answer if he answers "yes" to questions 1, 4, 6, 7, 9, 13, and 17, and if he answers "no" to questions 5, 8, 10, 12, 14, 16, 18, and 20. Then, the total number of points is calculated. Thus, the maximum number of points is 15. If the total score is more than 10, the respondent is altruistic; if it is less than 10, he is selfish.

*2.4. Analysis*

Models for a two-year combined sample. We encountered considerable difficulties when it came to retrieving the information based on the data of the experiments that were conducted as part of this research. Firstly, the data have discrete values (expressed as points), which results in an inevitable error within around one point. Secondly, the results of the tests are influenced by many factors not related to the way the learning process is organized: whether the student being tested feels well, whether he/she accidentally came across the vocabulary elsewhere, etc. Thus, the data appear to be quite noisy. The linear regression is most often used to build dependencies in problems of this kind. However, in this case, it is hardly applicable since the desired dependencies are highly nonlinear. The search for dependencies in the form of polynomials is complicated since the coefficients of the models are not resistant to the data errors. It means that the model does not greatly approximates the desired relationship as random measurement errors, which is particularly salient in the case of high-degree polynomials. In order to overcome the abovementioned obstacles, we processed the data with a tool which appears to be the most effective, i.e., neural networks [42].

We used the following neural network with one hidden layer:

$u_N(x) = \sum_{i=1}^{N} c_i v(x, a_i)$, where $v(x, a_i) = Tanh(a_i(x - b_i))$. The weights of the network $\{a_i, b_i, c_i\}$ were selected by minimizing the quadratic error function $\sum_{j=1}^{M} (u_N(x_j) - u_j)^2$.

## 3. Results

The graphs and charts below illustrate some of the results that we obtained. Figure 1 illustrates how the learning outcomes (the difference between the results of the final and the diagnostic tests) depend on the diagnostic test.

This graph shows that the outcomes in the exposure group are twice as high as those in the reference one. The difference is especially salient for the average students (sometimes it can be three times as high) because their collaboration with the top learners is remarkably fruitful. The decline in the learning outcomes associated with the higher results of the diagnostic test can be explained by the fact that high-achieving students cannot produce significantly better outcomes since their initial results were close to the highest possible score.

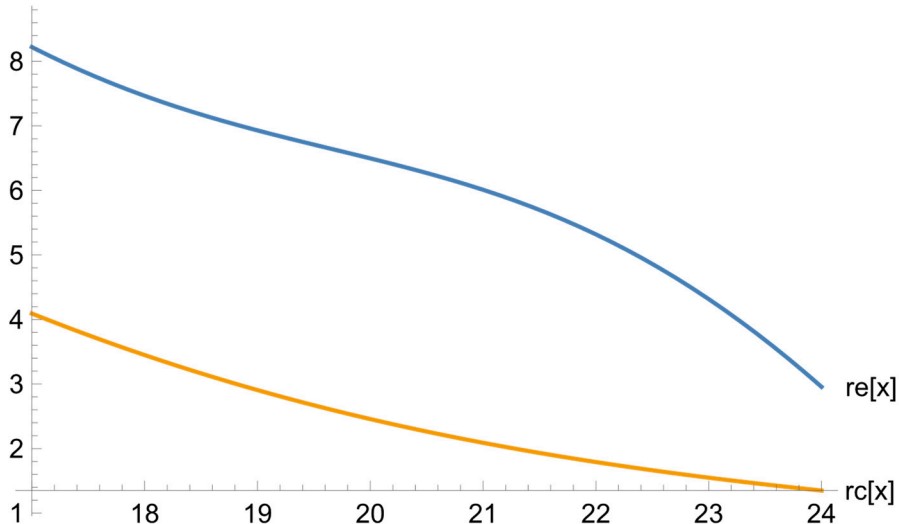

**Figure 1.** The dependence of the difference between the final and diagnostic tests on the diagnostic test for the reference and exposure groups.

Figure 2 illustrates the dependence of the psychological test results (the level of altruism) on the students' training level (on the basis of the diagnostic test results).

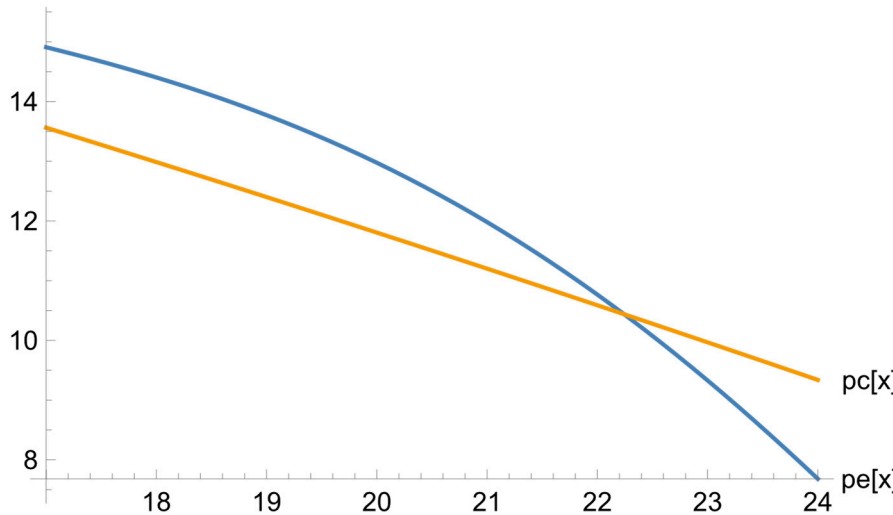

**Figure 2.** The dependence of the psychological test results on the diagnostic test results for the exposure and reference groups.

The graph shows that the higher the students' training level, the less altruistic they are. This may be caused by a free-rider approach typical of underachievers. In the exposure group, this downward trend is more marked. Since in the context of our research the subjects of the experiment act in the capacity of a teacher, high achievers have to give help to poorly performing learners.

Figure 3 illustrates how altruism influences learning achievement (after deducting the influence of the initial proficiency level). We consider the difference between the students' real scores and the model data. For instance, even if two students have the same diagnostic test scores, it does not necessarily mean that their final test results will be the same. The learners' performance is influenced by a multitude of factors, including psychological characteristics such as egoism or altruism. This dependence was built on the basis of the difference between the real data and the data corresponding to the dependences in Figure 1.

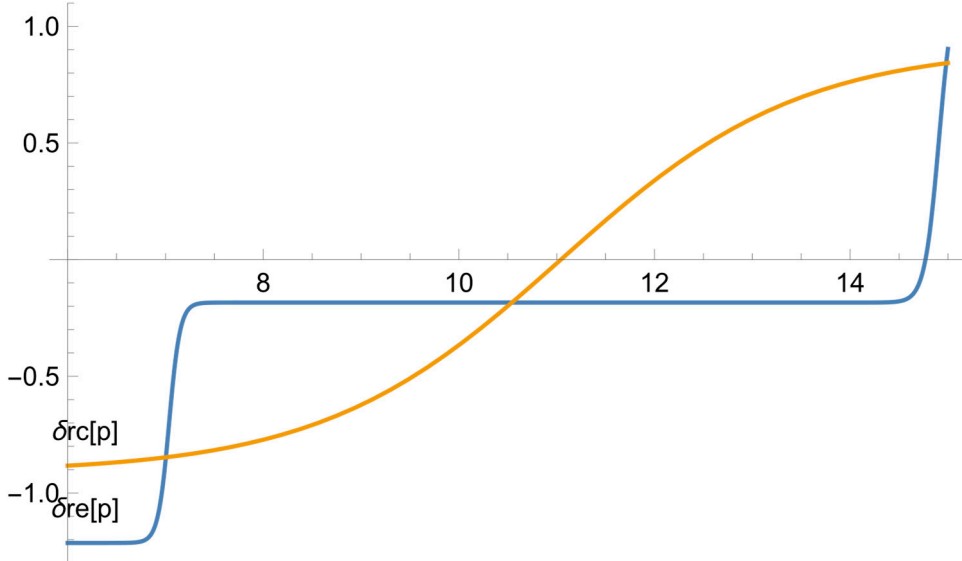

**Figure 3.** The dependence of the model error on the results of the psychological tests for the exposure and reference groups.

Both curves in this graph show that the dependence is quite small. The curve for the exposure group is smoother. It illustrates that the more altruistic learners' outcomes are somewhat better than those of the more selfish students, regardless of the way in which the learning process is organized. For the exposure group, the dependence is more complex. On the major part of the interval which shows the change in the results of the psychological test, there is virtually no dependence (the margin of change is within one point). Thus, for most of the students, their initial psychological setting hardly influences the learning outcomes. At the edges of the graph, the dependences are more pronounced. Thus, if the students are reluctant to help their peers, their psychological settings have an adverse effect on the learning outcomes. By the same token, if the students are enthusiastic about sharing their knowledge with the other learners, their positive attitude has a beneficial effect on their outcomes.

Then, we studied the dependence of the difference between the results of the final and diagnostic language tests in the reference and exposure groups on the results of the psychological test and the diagnostic linguistic test. This dependence makes it possible to consider simultaneously how both the initial level of foreign language training and the psychological characteristics influenced the learning outcomes.

To build the model, we used the following neural network with a single hidden layer:

$$u_N(x,y) = \sum_{i=1}^{N} c_i v(x,y;a_i).$$

The basis function is $v(x,y,a_i) = Tanh(a_i(x - b_i))Tanh(d_i(y - e_i))$. By the same token as previously stated, the weights of the network $\{a_i, b_i, c_i, d_i, e_i\}$ were selected by minimizing the quadratic error function $\sum_{j=1}^{M} (u_N(x_j, y_j) - u_j)^2$.

First and foremost, this approach enabled us to build the models with greater accuracy than in Figure 1. For the models in Figure 1, the maximum error of the model for the exposure group was 2.49 and the root-mean-square error was 1.13; the maximum error of the model for the reference group was 2.09 and the root-mean-square error was 0.89. For the model with two variables, the error of the model for the exposure group was 2.02 and the root-mean-square error was 0.79; the maximum error of the model for the reference group was 1.41 and the root-mean-square error was 0.481.

Figure 4 shows that at any values of the input variables (the results of the diagnostic and psychological tests), the results in the exposure group are better than those in the reference group. The maximum difference is achieved for the minimum scores for the diagnostic and psychological tests, and it accounts for over eight points.

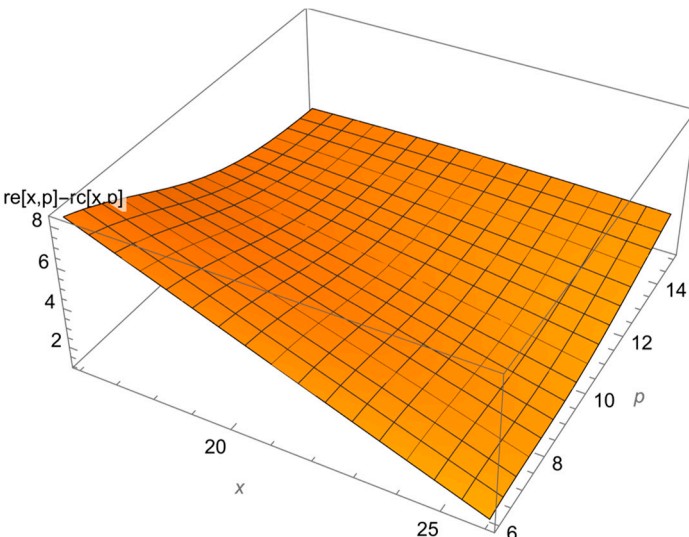

**Figure 4.** The difference between the results in the exposure and reference groups.

Figure 5 illustrate some two-dimensional sections which allow us to come to further conclusions (one variable is taken as a constant, and we consider its dependence on the other variable).

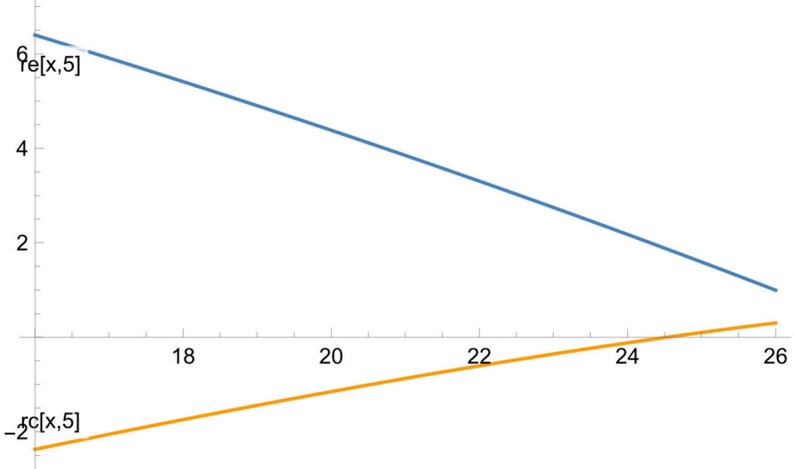

**Figure 5.** The dependence of the difference between the final and diagnostic linguistic tests for the exposure and reference groups on the diagnostic linguistic test at the minimum value of the psychological test.

Figure 5 illustrates the maximum difference between the dependences for the exposure and reference groups. The graph shows the dependence of the learning outcomes (the difference between the results of the final and diagnostic linguistic tests) at the minimum value of the psychological test. In the reference group, the results are negative, which means that the level of the students' knowledge is decreasing. Thus, if the students are uncooperative and averse to helping their peers in the course of collaborative learning, their knowledge shows no signs of improvement. In the exposure group, the situation is fundamentally different, and it is very similar to the overall dependence illustrated in Figure 1. Thus, the initially negative attitude to mutual support and assistance can be

changed by applying the experimental technique and has hardly any effect on the quality of training.

Figure 6 shows the same dependence as was illustrated in the previous figure but for the maximum value of the psychological test. Both of these dependences are quite similar; however, the exposure group curve is higher by a little more than three points. This suggests that the students who are willing to offer assistance helped each other in the reference group as well. Nevertheless, mutual assistance had a greater effect in the exposure group since it was purposely organized.

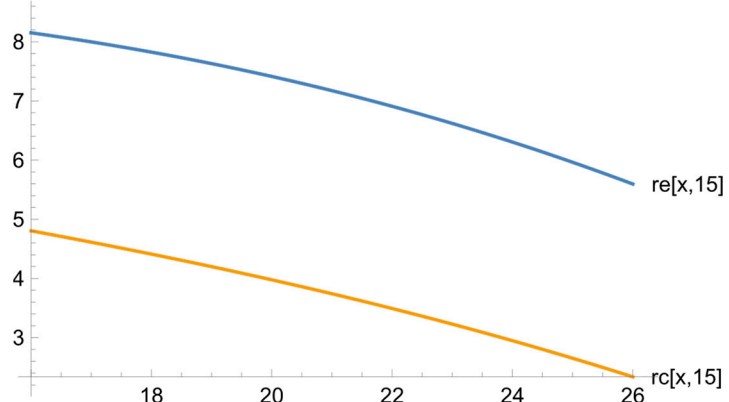

**Figure 6.** The dependence of the difference between the final and diagnostic linguistic tests for the exposure and reference groups on the diagnostic linguistic test at the maximum value of the psychological test.

Figures 7 and 8 show the dependences of the learning outcomes on the results of the psychological test, with the initial level of training fixed.

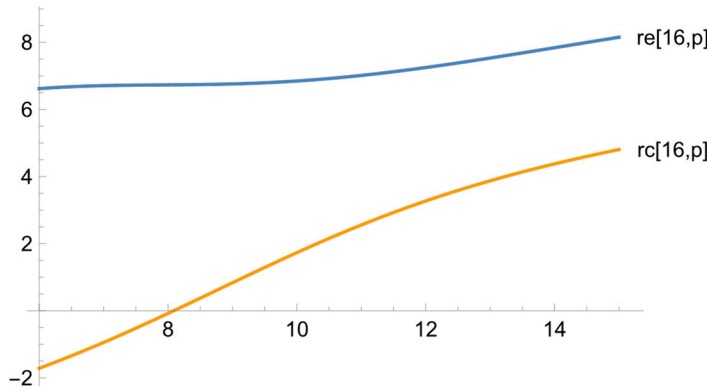

**Figure 7.** The dependence of the difference between the final and diagnostic linguistic tests for the exposure and reference groups on the psychological test at the minimum value of the diagnostic linguistic test.

For the students with the lowest levels of foreign language proficiency, the results in the exposure groups were considerably higher than those in the reference group, similarly to the other cases, but here, the dependences are of a fundamentally different nature. There is hardly any dependence for the exposure group, i.e., the results do not depend heavily on the initial psychological setting. However, for the reference group, the dependence is substantial. The knowledge of the students with a low level of foreign language proficiency who are averse to aiding their peers may not improve whatsoever. In the reference group, the students who are willing to help their peers can organize mutual assistance themselves, and the results will improve significantly, even though they will still be lower than those in the exposure group.

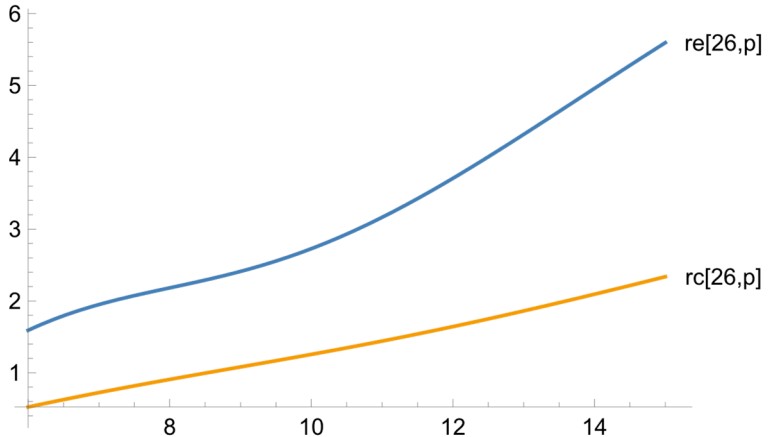

**Figure 8.** The dependence of the difference between the final and diagnostic linguistic tests for the exposure and reference groups on the psychological test at the maximum value of the diagnostic linguistic test.

For the students with high levels of foreign language proficiency, the results in both of the groups grow significantly, in direct proportion to the increase in altruistic attitudes, with the growth being more pronounced in the exposure group, in which the top-performing students who help the ones with low levels of foreign language proficiency learn more efficiently, especially if the learning process is geared towards such interaction.

Thus, the results of this experiment confirmed the hypothesis about the beneficial influence of peer teaching on the students' academic performance. The results of the diagnostic psychological and linguistic tests seemed to contradict the second hypothesis, according to which the levels of learners' altruism and their learning outcomes were positively correlated. The students with the higher training level appeared to be less altruistic than those whose results of the linguistic test were quite low. However, the results of the final psychological and linguistic tests confirmed the second hypothesis: the learning process in the exposure groups, which was geared at interaction, is therefore conducive to more altruistic and community-oriented behavior and is proven to contribute to better learning outcomes than in the reference groups, in which the teacher facilitated the classroom activities.

## 4. Discussion

In the current study, we evaluated the students' foreign language lexical and grammatical skills in the course based on the peer teaching methodology and analyzed the effect of their altruistic and egoistic behaviors on their learning results.

According to the results on learning outcomes, students improved their English proficiency in the exposure groups to a greater extent than in the reference groups, which confirms the adequacy of peer teaching approach usage for English learning purposes. At the same time, the results of the psychological test revealed that the higher the students' training level, the less altruistic they are. Also, it was detected that the more altruistic learners' progress in outcomes was higher than those of the more selfish students, regardless of the way in which the learning process was organized; this can be explained by the fact that for students who have higher level of English proficiency, it is harder to achieve substantial progress in comparison to students who have lower level of English. Moreover, the statistical data proved the efficiency of the peer teaching methodology for students' majoring in linguistics, despite their psychological characteristics. This methodology involved more egoistic students, who tend to have higher learning outcomes, for the purpose of helping and educating more altruistic students with lower learning outcomes. The effect was twofold: students with a higher level of English proficiency prepared carefully by reading additional material and making deeper analysis for the lessons, and as consequence, they improved their proficiency; students with a lower level of English

proficiency received processed material in a simple and understandable form for students with a high level of English and also improved their proficiency.

The results of this experiment closely correlate with the studies indicated in the theoretical section. First and foremost, the higher test scores in the exposure groups indicate that by teaching their peers, students acquire the learning techniques which can later be deployed in their further education as well as a more profound understanding of the subject [8]. Secondly, the students of the exposure groups worked in a more community-minded environment than the learners in the reference groups. By getting involved in peer mentorship, they were expected to display more altruistic behavior, and in so doing, they produced better results in their final test. Such outcomes clearly indicate that working for the benefit of society gives the whole group more chances of survival and contributes to its more considerable accomplishments [15]. The fact that the results of the linguistic tests grew in direct proportion to the increase in altruistic attitudes vividly illustrates that if all of the members of the group share "collective concepts of the good", it leads to a significant increase in their performance, since they work towards the same goal in a secure favorable environment [34]. Lastly, by assuming the role of mentors, the learners in the exposure groups more actively participated in the classroom interactions. Some of the students in the exposure groups, whose initial attitude to mutual support and assistance was negative, changed their perspective when the experimental technique was applied. It once again confirmed the idea put forward by S.R. Brown and M.R. Brown that active involvement in social interaction encourages altruistic behavior [39]. The theoretical work of other scholars in this field has been a useful resource for planning and designing, and we expect that our study will provide something of value for future researchers too.

This study was based on the work of scientists involved in the analysis of factors which influence the students' learning outcomes in EFL classes [6,42] and the relationship between altruistic and egoistic behaviors and students' motivation to learn [22]. In contrast to previous studies, our paper presents a valuable analysis on the relationship between the students' altruistic and egoistic behaviors (psychological factors) and their foreign language lexical and grammatical skills in two different settings—a traditional classroom and a course based on the peer teaching methodology.

The limitations of this study include the sample size, which was relatively small. Since it was conducted over only 2 years, we implemented such an educational method in the curricular of the course, and the duration of the course was only one semester. Moreover, since the study was conducted only in Russia, cultural differences may affect the results of students from other countries. Only students of the humanities, especially linguists, took part in the study.

**Author Contributions:** Conceptualization, A.K. and A.D.; methodology, D.T.; software, D.T.; validation, A.K. and E.S.; formal analysis, A.K.; investigation, A.D.; resources, A.D.; data curation, N.A.; writing—original draft preparation, A.K.; writing—review and editing, A.K.; visualization, A.D.; supervision, A.K.; project administration, A.D.; funding acquisition, N.A. All authors have read and agreed to the published version of the manuscript.

**Funding:** This research received no external funding.

**Institutional Review Board Statement:** This study was conducted in accordance with the Declaration of Helsinki and approved by the Institutional Review Board of the INSTITUTE OF HUMANITIES (protocol code 349, 16 February 2021).

**Informed Consent Statement:** Informed consent was obtained from all subjects involved in this study.

**Data Availability Statement:** The data presented in this study are available on request from the corresponding author. The data are not publicly available due to privacy restrictions.

**Conflicts of Interest:** The authors declare no conflict of interest.

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
