# Peer review of "Evaluation of Linguistics Students’ Learning Outcomes in Peer Teaching Courses: The Effect of Altruistic and Egoistic Behaviors"

_ejihpe, doi:10.3390/ejihpe13110185_

Round 1

Reviewer 1 Report

Comments and Suggestions for Authors Dear authors

-The title would be advisable to modify, seeking a less complex and more clarifying form about the center of the research.

The research questions and research objectives are not made explicit.

Nor is there an adequate identification of variables and possible hypotheses that may relate them. There is a kind of inertia that juxtaposes the question of the very diffuse didactic method and personality dimensions, without appreciating the research itinerary that can appreciate integrated solvent results to establish relationships, whether causal or only associative.
It would be very convenient, once the relevant issues indicated above are resolved, to differentiate between the discussion of results and conclusions, and for them to respond in a clear and synthetic manner to the questions and objectives (when these are defined.
The references must be reviewed and homogenized since some are incomplete or contain formatting errors, and an updated improvement of the background is recommended.

Regar

Author Response

Thank you very much for your valuable comments! Please, find the answers in the attached file.

Reviewer 2 Report

Comments and Suggestions for Authors

This study is interesting due to its novelty compared to previous research. It presents a comprehensive and rigorous statistical analysis that verifies the initial research questions. However, further elaboration is needed in discussing the data, especially in comparison to similar studies. This work primarily targets education professionals’ trainers, hence this last suggestion for improvement.

Author Response

(The authors gave the same response as above.)

Reviewer 3 Report

Comments and Suggestions for Authors

The article is particularly interesting due to the special approach of the development of language competences, its connection with personality traits and the impact on students’ involvement on learning outcomes. The originality of the article resides in presenting a very fruitful but still not common way to enhance students’ language skills: their direct involvement in the teaching process, and in the method used in analysing the data collected. However, from my perspective, the benefits for the use of neural networks in the analysis should be a bit further explained, as compared to other methods.

 In addition, the discussions could have been more extensive.

Author Response

(The authors gave the same response as above.)

Round 2

Reviewer 1 Report

Comments and Suggestions for Authors

Dear authors

The improvements made are important and satisfy the requirements of the previous review.

Regards

Author Response

Thank you!

Reviewer 2 Report

Comments and Suggestions for Authors

Thank you very much for your work. I have carefully read the justifications for the changes made to the study based on my suggestions, and I am satisfied with them. Only one new comment regarding the multiple-choice tests has been added for your consideration. Congratulations.

Author Response

Thank you! We added explanation about usage of multiple-choice tests. 
